# Heterologous Watermelon *HSP17.4* Expression Confers Improved Heat Tolerance to *Arabidopsis thaliana*

**DOI:** 10.3390/cimb47080606

**Published:** 2025-08-01

**Authors:** Yajie Hong, Yurui Li, Jing Chen, Nailin Xing, Wona Ding, Lili Chen, Yunping Huang, Qiuping Li, Kaixing Lu

**Affiliations:** 1Ningbo Key Laboratory of Agricultural Germplasm Resources Mining and Environmental Regulation, College of Science & Technology, Ningbo University, Ningbo 315300, China; 2311130014@nbu.edu.cn (Y.H.); 18987757278@163.com (Y.L.); 18458658237@163.com (J.C.); dwn@zju.edu.cn (W.D.); chenlili2@nbu.edu.cn (L.C.); 2School of Marine Sciences, Ningbo University, Ningbo 315211, China; 3Institute of Vegetables, Ningbo Academy of Agricultural Sciences, Ningbo 315040, China; xingnailin@hotmail.com (N.X.); hyp2003@163.com (Y.H.)

**Keywords:** watermelon, *HSP17.4* gene, heat stress

## Abstract

Members of the heat shock protein 20 (HSP20) family of proteins play an important role in responding to various forms of stress. Here, the expression of *ClaHSP17.4* was induced by heat stress in watermelon. Then, a floral dipping approach was used to introduce the pCAMBIA1391b-GFP overexpression vector encoding the heat tolerance-related gene *ClaHSP17.4* from watermelon into *Arabidopsis thaliana,* and we obtained *ClaHSP17.4*-overexpressing *Arabidopsis* plants. Under normal conditions, the phenotypes of transgenic and wild-type (WT) *Arabidopsis* plants were largely similar. Following exposure to heat stress, however, the germination rates (96%) of transgenic *Arabidopsis* plants at the germination stages were significantly higher than those of wild-type idopsis (17%). Specifically, the malondialdehyde (MDA) content of transgenic *Arabidopsis* was half that of the control group, while the activities of peroxidase (POD) and superoxide dismutase (SOD) were 1.25 times those of the control group after exposure to high temperatures for 12 h at the seedling stages. The proline content in *ClaHSP17.4*-overexpressing transgenic *Arabidopsis* increased by 17% compared to WT plants (* *p* < 0.05), while the soluble sugar content rose by 37% (* *p* < 0.05). These results suggest that *ClaHSP17.4* overexpression indirectly improves the antioxidant capacity and osmotic regulatory capacity of *Arabidopsis* seedlings, leading to improved survival and greater heat tolerance. Meanwhile, the results of this study provide a reference for further research on the function of the *ClHSP17.4* gene and lay a foundation for breeding heat-tolerant watermelon varieties and advancing our understanding of plant adaptation to environmental stress.

## 1. Introduction

China is the world’s largest producer and consumer of watermelons, with an annual production of 300 million tons, accounting for 70.6% of global output (Food and Agriculture Organization; 2023). Watermelon plants are often exposed to various unfavorable environmental conditions such as drought, temperature variations, and salt stress. Drought can cause dehydration of plant cells, osmotic imbalance, and reduced photosynthetic activity [1]. High- and low-temperature stress disrupts the structure and function of plant cells, resulting in decreased chlorophyll levels and impaired photosynthesis [2,3]. Excessive salt accumulation can lead to premature aging and wilting of leaves [4]. These abiotic stresses ultimately lead to a significant reduction in watermelon yield. To counteract these challenges, plants have evolved a series of defensive responses, many of which are characterized by changes in gene expression that enhance stress resistance [5].

Heat shock proteins (HSPs) are a group of evolutionarily conserved proteins that are upregulated in plants exposed to abiotic stress conditions [6]. These HSPs primarily function as molecular chaperones that confer resistance to heat and salinity, stabilizing plant cellular structures [6]. HSPs also serve an important regulatory role during plant development [7]. HSP family members are classified into five subfamilies based on their molecular weights and sequence similarity, including the HSP100, HSP90, HSP70, HSP60, and small HSP (sHSP/HSP20) families [7]. HSP20s range in size from 12 to 43 kD [8] and contain a highly conserved C-terminal α-crystallin domain (ACD), whereas their N-terminus tends to be more variable [9]. The ACD consists of a series of β-strands in the N-terminal common region I (CRI; β2-β3-β4-β5), a C-terminal common region II (CRII; β7-β8-β9), and a hydrophobic ring (β6-ring) linking these CRI and CRII domains CRI with CRII [10]. HSP20s rapidly accumulate in response to heat stress and can bind to denatured proteins, protecting against permanent protein aggregation. When the stress conditions have abated, HSP20 can then separate from these denatured proteins, allowing them to interact with additional molecular chaperones [11]. Under normal physiological conditions, HSP20s are generally undetectable in plant cells, although they are induced by stresses including oxidative stress, cold, drought, salinity, and biotic stress conditions whereupon they serve to help improve tolerance through the preservation of cellular integrity and homeostasis [10]. Constitutive HSP20 expression in transgenic plants has been shown to enhance stress resistance and productivity under various stress conditions [10]. For example, constitutively expressing *TaHSP17.6* in *Arabidopsis thaliana* improves salt tolerance and reduces sensitivity to exogenous ABA [12]. Heterologous *LimHSP16.45* expression enhances *Arabidopsis* viability under conditions of oxidative stress, high salt levels, and high temperatures, with heat shock granules (HSGs) forming in response to heat or salt stress [13]. Overexpressing *SIHSP17.7* has been reported to increase tolerance to cold stress in tomatoes by promoting sucrose accumulation within cells while mitigating reactive oxygen species (ROS) production [14]. In rice, HSP20 overexpression has similarly been reported to improve resistance to dehydration, drought, and ultraviolet-B radiation exposure [15]. Transgenic *Arabidopsis* plants expressing *MsHSP16.9* and *TaHSP23.9* also exhibit more robust growth and superior tolerance when exposed to high temperatures [16]. Overexpressing HSP20 in *LimHSP16.45* in the *Arabidopsis hsp17.6II* mutant line has also been shown to increase abiotic stress tolerance [17]. However, research on HSP genes in watermelon (*Citrullus lanatus*) remains limited. To date, only one study has provided valuable insights into the classification and expression patterns of watermelon HSP genes [18], but it did not elucidate the specific functional roles of individual HSP genes under heat stress. This gap in knowledge highlights the limitations of current research on watermelon HSP genes, particularly the lack of functional characterization of specific HSP members. Given the economic importance of watermelon as a globally cultivated crop and its susceptibility to high-temperature stress, understanding the molecular mechanisms underlying its thermotolerance is of great significance.

In this context, our study focuses on the functional analysis of the watermelon *HSP17.4* (ClCG04G005380) gene in response to high-temperature stress. *ClaHSP17.4* (ClCG04G005380) is one of the three genes most significantly upregulated among the 44 HSP genes in watermelon under heat stress at 42 °C. Furthermore, the promoter region of *ClaHSP17.4* contains cis-elements responsive to heat stress [18]. *ClHSP17.4* may thus play an important role in watermelon heat stress responses. In this study, an effort was made to better understand the functions of the *ClaHSP17.4* gene. To that end, *ClaHSP17.4* was cloned from the watermelon variety 8424 and used to construct the pCAMBIA1391b-GFP overexpression vector, which was then introduced into *Arabidopsis* via an *Agrobacterium*-mediated transformation system. Using this approach, the impact of *ClaHSP17.4* on heat tolerance was evaluated, providing an evidence-based foundation for further studies of watermelon HSPs while also establishing a resource for future efforts to breed stress-resistant watermelon varieties.

## 2. Materials and Methods

### 2.1. Watermelon Growth and Treatment

Gene expression analyses were performed with the ‘8424’ watermelon variety. *Citrullus lanatus* cv. 8424 is known to be particularly sensitive to abiotic stresses, including high temperature, which is also often used in investigating the molecular mechanisms underlying heat tolerance [19,20]. These plants were grown in nutrient substrate (https://e.tb.cn/h.hQzXpsTisAsBPdk?tk=fkIW4diPAoa CZ005 (accessed on 12 June 2022)) for watermelon cultivation (pH6.5) in a temperature-controlled greenhouse with daytime/nighttime temperatures of 28/22 ± 1 °C, a light intensity of 200 µmol·m^−2^·s^−1^, and a 16 h photoperiod. Three-week-old watermelon seedlings were exposed to exogenous heat stress (42 °C), collecting leaves after 0, 1, 4, and 12 h of treatment in a growth chamber (Yanghui Instrument Co. Ltd., Ningbo, China) [21]. Three biological replicates were performed for each treatment and contained 15 seedlings. All materials were immediately frozen in liquid nitrogen and stored at −80 °C until RNA isolation.

### 2.2. Bioinformatics Analysis

Conserved domain prediction was performed using the SMART online software (http://smart.embl-heidelberg.de/ (accessed on 12 February 2022)). Analysis of protein molecular weight and theoretical isoelectric point was completed using ProtParam-EXPASY (https://web.expasy.org/protparam/ (accessed on 12 February 2022)). Subcellular localization prediction analysis was conducted via the online tool WoLF-PSORT (https://wolfpsort.hgc.jp/ (accessed on 12 February 2022)). Signal peptide analysis was performed with SignalP4.1 (http://www.cbs.dtu.dk/services/SignalP/ (accessed on 12 February 2022)). Protein hydrophobicity analysis was carried out using the EXPASY tool. Transmembrane domain prediction of the protein was achieved via TMHMM SERVER (https://services.healthtech.dtu.dk/services/TMHMM-2.0/ (accessed on 12 February 2022)). The promoter elements of *ClaHSP17.4* were analyzed using the Plantcare online promoter analysis software (http://bioinformatics.psb.ugent.be/webtools/plantcare/html/ (accessed on 12 February 2022)).

### 2.3. ClaHSP17.4 Gene Collection and ClaHSP17.4: pCAMBIA1391b-GFP Overexpression Vector Construction

The *ClaHSP17.4* gene sequence was downloaded from the *Cucurbitaceae* genome database (http://www.cucurbitgenomics.org/ (accessed on 4 February 2022)) and genome version (e.g., Charleston Gray) on 11 March 2022, and corresponding primers were developed with Primer Premier 5.0. Total RNA was then extracted from the developing leaves of ‘8424’ watermelons, followed by the synthesis of cDNA using the HiScript II 1st Strand cDNA Synthesis Kit (Sangon Biotech, Shanghai, China). qPCR was next used to amplify *ClaHSP17.4* using specific primers qPCR-ClaHSP17.4 (Table 1), after which TA cloning, transformation, screening, enzyme digestion, and sequencing were used to confirm that the *ClaHSP17.4*: pMD-19-T cloning vector had been successfully established. The qPCR reaction mix (50 μL) consisted of 25 μL 2× Taq Master Mix, 4 μL cDNA (~50 ng), 19 μL ddH_2_O, and 2 μL primers. The qPCR thermocycler settings were as follows: 94 °C for 4 min; 30 cycles of 94 °C for 4 min, 58 °C for 30s, 72 °C for 1 min, final extension at 72 °C for 5 min, and storage at 16 °C. Double-enzyme digestion of the *ClaHSP17.4*: pMD19-T cloning vector and the pCAMBIA1391b-GFP vector was then performed with *Kpn* I and *Bam*H I (with K buffer for 2 h in 37 °C), followed by recovery of the digestion products with the Universal DNA Purification Kit DP214 (TIANGEN, Beijing, China). The vectors were subjected to T4 ligation (T4 DNA Ligase, 16 °C overnight). The resultant *ClaHSP17.4*: pCAMBIA1391b-GFP constructs were sequenced by Wuhan Gene Create Bioengineering Co., Ltd. (Wuhan, China).

### 2.4. Transgenic ClaHSP17.4-Overexpressing Arabidopsis Plant Selection

An *Agrobacterium*-mediated floral soaking approach and antibiotic screening were generated using *ClaHSP17.4*: pCAMBIA1391b-GFP transgenic *Arabidopsis* plants. *Arabidopsis* plants were grown to the flowering stage in a growth chamber (Yanghui Instrument Co. Ltd., Ningbo, China) under controlled conditions: daytime/nighttime temperatures of 23/20 ± 1 °C, a light intensity of 135 µmol·m^−2^·s^−1^, and a 16 h light/8 h dark photoperiod.

All opened flower buds and developing siliques were removed before infiltration. The *Agrobacterium* culture containing the *ClaHSP17.4*: pCAMBIA1391b-GFP vector was collected and diluted with a 5% sucrose solution (containing 0.02% Silwet) to an OD_600_ of approximately 0.8–1.0. The *Arabidopsis* inflorescences were immersed in the suspension for 1–2 min, then covered with plastic wrap and incubated in darkness for 3 days before resuming normal cultivation. When a new batch of flower buds emerged (typically after one week), the infiltration process was repeated.

T_0_ seeds were collected and sown on MS medium supplemented with hygromycin (25 mg·L^−1^) to screen for hygromycin-resistant *Arabidopsis* plants. Robust seedlings were selected and transplanted into a growth substrate for further cultivation. Upon seed maturation, T_1_ seeds were collected individually from each plant. The hygromycin-resistant *Arabidopsis* lines were continuously cultivated to obtain homozygous T_3_ seeds, which were used for subsequent experiments and phenotypic analysis. DNA from these transgenic *Arabidopsis* plants was extracted, and positive lines were identified using gene-specific primers qPCR-ClaHSP17.4 (Table 1).

To detect the expression levels of *ClaHSP17.4* genes in *ClaHSP17.4*-overexpressing *Arabidopsis* lines, a LightCycler^®^ 96 instrument (Roche, Mannheim, Germany) and ChamQ Universal SYBR qPCR Master Mix (Vazyme, Nangjing, China) were used for qRT-PCR analyses, with β-*actin* (gene ID: Cla007792) serving as a normalization control [22]. Three biological replicates and a minimum of three technical replicates were used for each experiment. The primers used for these analyses are presented in Table 1. The qRT-PCR thermocycler settings were as follows: 95 °C for 10 min; 40 cycles of 95 °C for 5 s and 60 °C for 34 s. The primers are shown in Table 1. Relative target gene expression was computed with the 2^−ΔΔCt^ method and statistically analyzed via t-tests.

### 2.5. Heat Stress Treatment and Stress-Related Analyses

The experiment was conducted using a Completely Randomized Design (CRD) with three treatments and a control group. Each treatment was randomly assigned to experimental units to minimize bias. A total of 50 seeds from the T_3_ transgenic plants (*OE1*, *OE2*, and *OE3*, corresponding to different transgenic families) and wild-type (WT) plants were inoculated on 1/2 MS solid medium, establishing three replicates per group. Control seeds were cultivated at 22 °C, with a 16/8 (D/N) light cycle, a light intensity of 135 μmol·m^−2^·s^−1^, and relative humidity of 66%. Seeds on 1/2 MS solid medium in the high-temperature group were instead incubated for 24 h at 38 °C with a 16/8 (D/N) light cycle, a light intensity of 135 μmol·m^−2^·s^−1^, and relative humidity of 66%, before transfer to normal culture conditions. All seeds were then cultivated for an additional week, counting the numbers of germination events and cotyledons in each group while also recording phenotypes. The germination rate was assessed as follows: radicle emergence ≥1 mm, The survival rate was assessed as follows: green cotyledons post-recovery. Root length was assessed using ImageJ 1.48v analysis, according to n = 10 seedlings/replicate.

Seeds from the established *Arabidopsis* lines were surface-sterilized with 5% NaClO for 5 min, then washed 5 times with sterile distilled water. They were sown, and uniform seedlings with 5 ± 0.3 mm hypocotyl length and fully opened cotyledon expansion were selected on day 10 of growth for transfer to 1/2 MS solid medium and allowed to grow for 5 days before initiating stress experiments. Both control and high-temperature treatment groups were set up, with three replicate seedlings per group. Plants in the high-temperature stress group were treated for 24 h in a 38 °C incubator. Whole seedlings were sampled after 0 and 12 h of exposure to high-temperature conditions for analyses of malondialdehyde (MDA) content, superoxide dismutase (SOD) activity, and peroxidase (POD) activity. POD activity was measured as described by Beers and Sizer [22], while SOD activity was determined according to the method of Giannopolitis [23], and the MDA content was measured as described by Buege and Aust [24]. Proline and soluble sugar content were determined using the respective Solarbio detection kits (Beijing). DPS v7.05 was used to compare different traits with analyses of variance (ANOVAs).

## 3. Results

### 3.1. ClaHSP17.4 Gene Characteristics

The *ClaHSP17.4* (ClCG04G005380) open reading frame is 468 bp long and includes no intron, as shown in the watermelon genome database. The 468 bp *ClaHSP17.4* coding sequence was found to encode a 156-amino-acid protein with a predicted molecular weight of 17.4 kDa, according to the ProtParam tool, leading to the naming of *ClaHSP17.4* accordingly. The Pfam website predicted the presence of a C-terminal conserved Alpha crystallin/Hsp20 domain in ClaHSP17.4 identified with SMART. ClaHSP17.4 was predicted to include an Alpha crystallin/HSP20 domain spanning from amino acids 40 to 156, with no N-terminal signal peptide. This absence of a signal peptide was also confirmed using the online analysis tool SignalP4.1. The presence of a non_cytoplasmic_domain was indicative of a protein lacking a trans-membrane structural region, consistent with results from the TMHMM database. The predicted isoelectric point of ClaHSP17.4, as calculated with ProtScale, was 5.93, consistent with ClaHSP17.4 being an acidic protein. The measured instability index of 30.9 was suggestive of a stable protein, as the value was less than 40. Heat shock elements (HSEs) are cis-acting elements often found in HSP20 promoters, enabling the induction of gene expression in response to heat shock conditions. Additionally, predictions from the Wolf Psort database indicated that ClaHSP17.4 may be localized in the cytoplasm. The predicted *ClaHSP17.4* promoter contained many different cis-regulatory elements, including common eukaryotic transcriptional elements, elements responsive to hormones including methyl jasmonate, elements related to anaerobic conditions, and auxin-responsive elements, which are related to stress and defense responses (Table 2). Together, these findings indicate that *ClaHSP17.4* is an HSP20 family member that may be involved in watermelon stress and defense responses.

### 3.2. Analysis of ClaHSP17.4 Expression in Response to Heat Stress in Watermelon

Next, qPCR analyses were conducted, which revealed that *ClaHSP17.4* was highly upregulated in response to high-temperature stress in watermelon. A marked increase in *ClaHSP17.4* expression was evident at just 1 h after heat treatment, while after 4 h the expression levels of this gene were 20 times higher than those in control samples, although a decline in *ClaHSP17.4* expression was noted by 12 h (Figure 1).

### 3.3. Transgenic ClaHSP17.4-Overexpressing Arabidopsis Preparation

After extracting total RNA from watermelon leaves and reverse transcribing it to prepare cDNA, *Bam*H I and *Kpn* I restriction sites were introduced at the ends of *ClaHSP17.4* gene primers and used to amplify a 468 bp fragment (Figure 2A), which was then ligated into the pMD19-T cloning vector. Enzymatic digestion results for the positive recombinant vector are shown in Figure 2B. The *ClaHSP17.4* gene was inserted into the pCAMBIA1391b-GFP-overexpressing vector via enzymatic digestion, and recovery and enzymatic digestion were then used to confirm that the recombinant *ClaHSP17.4*: pCAMBIA1391b-GFP plasmid had been successfully constructed, yielding a ~468 bp band (Figure 2C). Plasmid sequencing confirmed the presence of the *ClaHSP17.4* gene in the forward orientation within this *ClaHSP17.4*: pCAMBIA1391b-GFP vector. The resultant vector was then used to transform competent *Agrobacterium* cells, followed by their use to infect *Arabidopsis* using a floral dip method. Transgenic *Arabidopsis* plants overexpressing *ClaHSP17.4* (*ClaHSP17.4*-OE) were selected with hygromycin, producing a limited number of T1 seeds with normal growth (Figure 3A). PCR analyses were used to confirm the presence of the *ClaHSP17.4* gene in the resultant plants (Figure 3B), confirming successful *ClaHSP17.4* integration into the *Arabidopsis* genome. *ClaHSP17.4*-OE lines OE1, OE2 and OE3 were the plant materials used for subsequent experiments. To confirm the successful overexpression of *ClaHSP17.4* in the three transgenic *Arabidopsis* lines, quantitative reverse transcription PCR (qRT-PCR) was performed to measure the expression levels of *ClaHSP17.4*. The results, as illustrated in Figure 3C, indicate that the expression levels of *ClaHSP17.4* were significantly elevated in the transgenic lines compared to the wild-type *Arabidopsis*. Specifically, the expression levels in *ClaHSP17.4*-OE-1, *ClaHSP17.4*-OE-2, and *ClaHSP17.4*-OE-3 were found to be approximately 56,656, 45,084, and 47,069 times higher, respectively, than in the control line. These findings confirm that *ClaHSP17.4* was successfully overexpressed in these transgenic lines, validating the effectiveness of the genetic modification.

### 3.4. Heterologous ClaHSP17.4 Expression Enhances Arabidopsis Heat Tolerance at the Germination Stage

No significant differences in germination rates were noted when comparing T3 *ClaHSP17.4*-OE and WT *Arabidopsis* under normal conditions (Figure 4A), with both varieties exhibiting respective 95% germination rates after cultivation for 7 days (Figure 4B). Upon exposure to high-temperature stress, the germination rates for transgenic *ClaHSP17.4*-OE were significantly higher than those for WT controls (Figure 4C). Specifically, the transgenic germination rates were 96%, as compared to just 17% for WT *Arabidopsis* under heat stress (38 °C) (Figure 4D). The *ClaHSP17.4* gene can thus enhance *Arabidopsis* heat tolerance during the germination stage.

### 3.5. ClaHSP17.4 Improves Arabidopsis Seedling Heat Tolerance

When comparing the WT and *ClaHSP17.4*-overexpressing transgenic *Arabidopsis* plants from the T_3_ generation, no significant differences in root length or cotyledon morphology were observed under normal conditions (Figure 5A). However, heat stress elicited less severe damage in the transgenic seedlings relative to WT *Arabidopsis* controls (Figure 5B). Moreover, the cotyledons of transgenic *Arabidopsis* seedlings were stronger and brighter colored as compared to those of WT plants. Transgenic plants also exhibited longer roots (0.5–1 cm) than WT controls under high-temperature stress, with a better-developed root system (Figure 5B,C). The *ClaHSP17.4* gene thus appears to enhance *Arabidopsis* seedling heat tolerance.

### 3.6. Impact of Heat Stress on MDA Content, Soluble Sugar, Proline Content, SOD Activity, and POD Activity in Arabidopsis Seedlings Overexpressing ClaHSP17.4

Lastly, the impact of heat stress on MDA levels, SOD activity, and POD activity was analyzed in WT and *ClaHSP17.4*-overexpressing transgenic *Arabidopsis* (Figure 5). Heat stress exposure was associated with lower levels of SOD and POD activity in WT *Arabidopsis* relative to the levels in transgenic plants, with some time-dependent fluctuation. These activity levels were persistently lower in WT *Arabidopsis* relative to the transgenic plants (Figure 5D,E), whereas the opposite was true with respect to MDA levels (Figure 5F). Specifically, the MDA content of transgenic *Arabidopsis* was half that of the control group, while the activities of POD and SOD were 1.25 times those of the control group after exposed to high temperatures for 12 h at the seedling stages. These findings demonstrate the ability of *ClaHSP17.4* to indirectly enhance antioxidant responses in *Arabidopsis*, ultimately contributing to improved heat stress. The changes in osmotic adjustment substances under heat stress are shown in Figure 5G,H. Under normal conditions, there was no significant difference in proline and soluble sugar content between the wild-type and overexpressed lines (*p* > 0.05). After 12 h of high-temperature stress, the content of both osmotic adjustment substances increased significantly, with the accumulation rate of proline and soluble sugars in the overexpressed lines being notably higher than in WT *Arabidopsis*. The proline content in *ClaHSP17.4*-overexpressing transgenic *Arabidopsis* increased by 17% compared to WT (* *p* < 0.05) (Figure 5G), while the soluble sugar content rose by 37% (* *p* < 0.05) (Figure 5H). These data indicate that overexpression of the *ClaHSP17.4* gene can significantly promote the synthesis and accumulation of osmotic adjustment substances under high-temperature stress, thereby enhancing the plant’s heat tolerance.

## 4. Discussion

The increasingly common outbreaks of extreme weather in recent years have contributed to declining watermelon quality and yields, adversely affecting overall watermelon production efforts. It is thus vital that the tolerance of watermelon crops to heat stress be improved, with the use of heat-resistant watermelon germplasm resources for the molecular breeding of stress-resistant varieties representing a particularly promising approach. To that end, the identification and characterization of stress resistance-related genes is particularly important. Members of the HSP20 family of proteins are particularly important mediators of stress tolerance that are involved in cell structural maintenance, protein folding, and the preservation of homeostatic balance in response to a range of environmental challenges [25]. In this study, the temperature response-related *ClaHSP17.4* gene encoding a 211-amino-acid protein was successfully cloned from the ‘8424’ watermelon variety. Sequence analysis revealed that ClaHSP17.4 contains a conserved C-terminal ACD_sHs-like domain, a hallmark of the HSP20 family. This domain functions by binding misfolded proteins to prevent irreversible aggregation rather than refolding them. The N-terminus of ClaHSP17.4 is involved in binding denatured proteins, while its C-terminus facilitates homo-oligomerization and the formation of heat stress particles, which are crucial for thermotolerance. These features suggest that *ClaHSP17.4* acts as a molecular chaperone, participating in oligomerization processes to protect cells under heat stress [26]. Whole-genome identification results led to the assignment of *ClaHSP17.4* to the CII subgroup of HSP20s [18], which primarily localize to the cytosol and nucleus [18]. The assignment of ClaHSP17.4 to this CII subgroup is supported by the presence of a conserved ACD domain and its nucleus localization. Members of this CII subfamily are evolutionarily conserved and responsive to heat stress in plants [25], that is correlates with the role of the chaperone αB-crystallin polydispersity in maintaining lens transparency while avoiding crystallization [27]. Together with HSP101, CII HSP20s have been reported to be essential for recovery following exposure to severe heat stress (10 h, 45 °C) while also exhibiting independent functions in *Arabidopsis* [28]. These findings suggest that *ClaHSP17.4* likely contributes to thermotolerance under high-temperature conditions. To further explore its regulatory mechanisms, we analyzed the promoter region of *ClaHSP17.4* and its expression patterns under heat stress. The promoter was found to contain multiple stress- and defense-related cis-acting elements (Table 2 and Figure 1), and transcript levels of *ClaHSP17.4* were significantly upregulated in response to heat stress. This is consistent with previous studies, such as the induction of *LrHSP17.2* expression under heat stress and the enhanced thermotolerance observed in *LrHSP17.2*-overexpressing plants [29]. These findings suggest that *ClaHSP17.4* may be involved in plant heat resistance regulation, which is consistent with the stress-responsive mechanisms of HSP gene family members in various plant species.

HSP20 expression is observed as a component of the developmental program engaged during seed maturation [30]. Class II HSP20s may have endowed early land plants with desiccation tolerance, while also conferring similar benefits to seeds. Additionally, HSPs are implicated in translational reactivation during seed imbibition, as evidenced by their interactions with specific translation factors under heat stress conditions [28,30,31]. This suggests that HSPs not only protect cellular structures but also facilitate the resumption of metabolic activities essential for germination. Supporting this notion, seeds overexpressing *PpHSP20-32* exhibit enhanced thermotolerance when exposed to high temperatures [32], highlighting the functional importance of HSP20s in stress adaptation. In this study, we investigated the physiological role of *ClaHSP17.4* in plant heat stress responses by overexpressing this gene in *Arabidopsis*. Our results demonstrate that plants overexpressing *ClaHSP17.4* exhibit significantly improved heat stress tolerance during the bud stage compared to wild-type plants (Figure 4). This enhanced thermotolerance likely stems from the ability of *ClaHSP17.4* to stabilize cellular proteins and maintain cellular homeostasis under high-temperature conditions. Furthermore, the overexpression of *ClaHSP17.4* may improve *Arabidopsis* germination rates under high temperatures, suggesting its potential role in safeguarding reproductive success in heat-stressed environments. These findings underscore the physiological significance of *ClaHSP17.4* as a key player in heat stress responses, with implications for improving crop resilience to climate change.

In response to environmental stress conditions, HSP20s can alter their structural features to limit harmful compound formation, exert antioxidant effects, promote cellular homeostasis, improve plant survival, and confer better resistance under a range of noxious conditions [10]. Specifically, cytosolic HSP20s are closely associated with plant heat resistance [33]. Similarly, the pepper *CaHSP16.4* gene has been shown to be associated with improved heat tolerance and survival in transgenic plants, attributable to a reduction in electrical leakage [34]. *PtHSP17.8* overexpression in *Arabidopsis* has been reported to be associated with improved health and salt stress tolerance [35]. Overexpression of *CaHsp25.9* enhances thermotolerance in pepper [36]. In this study, *ClaHSP17.4* overexpression similarly enhanced *Arabidopsis* seedling tolerance to heat stress such that these transgenic seedlings grew better than WT controls under high-temperature conditions, with corresponding improvements in the number of lateral roots and root length. This suggests that root characteristics are shaped by *ClaHSP17.4* under high-temperature stress in a manner that allows plants to more effectively adapt to these conditions. This aligns with previous findings that robust lateral root development and root hair proliferation are key adaptive responses to drought and heat [37].

Under abiotic stress conditions, plants can eliminate ROS under these conditions through the enhancement in SOD, POD, and CAT activity. ROS biogenesis can also trigger unsaturated membrane fatty acid peroxidation, leading to MDA production [38]. When MDA accumulates at high levels, this can cause cell membrane damage such that MDA content is often analyzed as a measure of membrane damage in stress-exposed plants. For example, *CmHSF30* transgenic *Arabidopsis* showed marked thermotolerance enhancement under heat stress, attributed to enhanced SOD activity, and reduced accumulation of MDA [39]. Similarly, *LrHSP17.2* transgenic plants significantly increased the activities of CAT, SOD, and POD under stress and decreased the content of MDA to enhance tolerance under heat stress [29]. In this study, *ClaHSP17.4* overexpression in transgenic *Arabidopsis* was associated with significant increases in SOD and POD activity under heat stress conditions relative to those in WT plants, while also reducing MDA accumulation (Figure 5D–F). These findings suggest that *ClaHSP17.4* enhances antioxidant capacity, protecting cellular membranes from oxidative damage and improving seedling survival under high-temperature stress. Transgenic *Arabidopsis* seedlings thus appear to improve antioxidant and osmoregulatory activity under high-temperature stress conditions, leading to better survival and stress tolerance at the seedling stage.

Under abiotic stress conditions, the accumulation of a large number of harmful chemicals will lead to an imbalance in the osmotic pressure of plant cells, and the structure and function of the cell membrane will be impaired; in this case, plants will use their metabolic pathways to accumulate a large number of osmotic regulatory substances (such as proline, soluble sugar, betaine, etc.), which will help maintain the osmotic pressure of the cell and protect the system components of the cell membrane to minimize the harmful effects of stress [40,41]. Overexpression of *Nicotiana tabacum Hsp17.6* enhances abiotic stress tolerance in Brassica napus by regulating the content of osmoregulatory substances, such as proline and soluble sugars, thereby improving the plant’s osmotic adjustment capacity [41]. Similarly, our study demonstrates that *ClaHSP17.4* overexpression in transgenic *Arabidopsis* significantly enhances heat tolerance through a similar mechanism. Under normal growth conditions, there was no significant difference in the levels of soluble sugars and proline between the wild-type and overexpressed lines. However, after 12 h of high-temperature stress, the proline content in *ClaHSP17.4*-overexpressing transgenic *Arabidopsis* increased by 17% compared to WT (* *p* < 0.05) (Figure 5G), while the soluble sugar content rose by 37% (* *p* < 0.05) (Figure 5H). The increased accumulation of proline and soluble sugars likely stabilizes cellular structures, protects enzymes and membranes, and maintains osmotic balance under heat stress, thereby mitigating cellular damage and improving plant survival [41]. These findings suggest that *ClaHSP17.4*-overexpressing transgenic *Arabidopsis* may enhance its heat tolerance by modulating the levels of osmoregulatory substances, such as proline and soluble sugars, thereby improving the osmotic adjustment capacity of the overexpressing line.

The current study primarily focused on the physiological and molecular responses of *ClaHSP17.4*-overexpressing plants under high-temperature stress. However, the underlying regulatory mechanisms, particularly the interactions between *ClaHSP17.4* and other stress-responsive genes or pathways, remain to be fully elucidated. This study was limited to *Arabidopsis thaliana* as a model organism, and further validation in crop species is necessary to assess the translational potential of *ClaHSP17.4* for agricultural applications. We will focus on performing transgenic experiments in watermelon to directly investigate *ClaHSP17.4*’s effect on fruit formation under heat stress. We will analyze the morphological, physiological, and molecular changes in fruit development caused by *ClaHSP17.4* overexpression or knockout under heat stress, and compare the results with those obtained in the current model system to provide a comprehensive understanding of the gene’s function. Meanwhile, field trials should be conducted to evaluate the performance of *ClaHSP17.4*-overexpressing watermelon under natural environmental conditions, including combined stresses such as drought and heat. Functional validation of *ClaHSP17.4* in economically important crop species, such as rice, wheat, and maize, will be essential to determine its potential for enhancing crop resilience to climate change.

## 5. Conclusions

*ClaHSP17.4* plays a role in high-temperature stress responses and is capable of promoting the accumulation of osmoregulatory substances and antioxidant activity in leaves (Figure 6), which was manifested in the increase in proline content of 17%, the increase in soluble sugar content of 37%, and the increase in POD and SOD activities of 25%. Meanwhile, *ClaHSP17.4* reduces membrane lipid peroxidation and cell membrane permeability. Therefore, *ClaHSP17.4*-overexpressing transgenic *Arabidopsis* is resistant to heat at the germination and seedling stages, and the germination rate of *ClaHSP17.4*-overexpressing transgenic *Arabidopsis* under heat stress is not affected by high temperature.

## Figures and Tables

**Figure 1 cimb-47-00606-f001:**
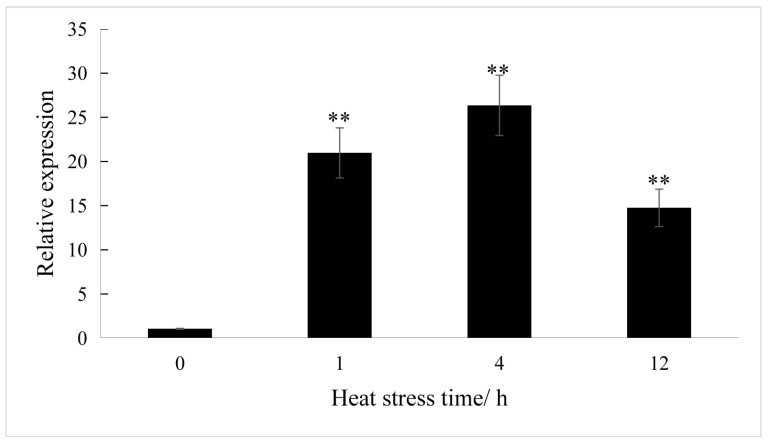
Relative expression level of *ClaHSP17.4* gene under heat stress. Note: **: very significant difference, *p* ≤ 0.01.

**Figure 2 cimb-47-00606-f002:**
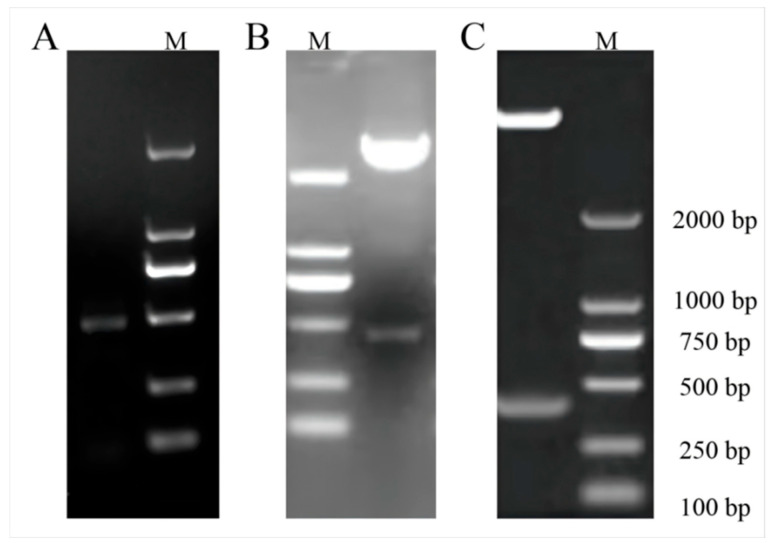
Construction of *ClaHSP17.4* gene overexpression vector. (**A**) PCR amplification of *ClaHSP17.4*; (**B**) double digestion of *ClaHSP17.4*-pMD19-T; (**C**) double digestion of *ClaHSP17.4*-pCAMBIA1391b-GFP. M: DL2000 DNA Marker.

**Figure 3 cimb-47-00606-f003:**
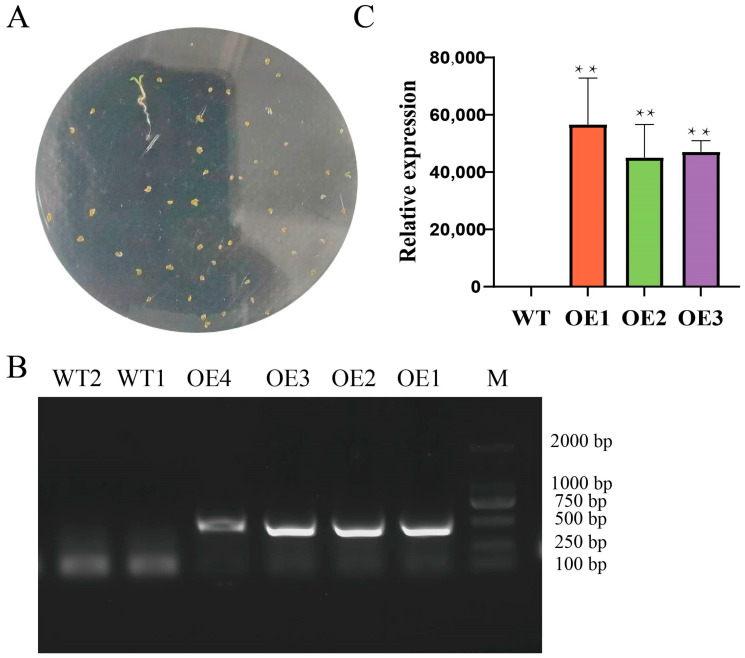
Confirmation of *ClaHSP17.4*-overexpressing *Arabidopsis* lines. (**A**) Part of results of hygromycin screening of *ClaHSP17.4*-overexpressing *Arabidopsis* lines; (**B**) PCR confirmation of *ClaHSP17.4*-overexpressing *Arabidopsis* lines using genomic DNA as template; (**C**) Gene expression levels in *ClaHSP17.4*-overexpressing *Arabidopsis* lines. M: DL2000 DNA Marker. WT1, WT2: wild-type lines. OE1, OE2, OE3: *ClaHSP17.4*-overexpressing *Arabidopsis* lines. Note: **: significant difference, *p* ≤ 0.01.

**Figure 4 cimb-47-00606-f004:**
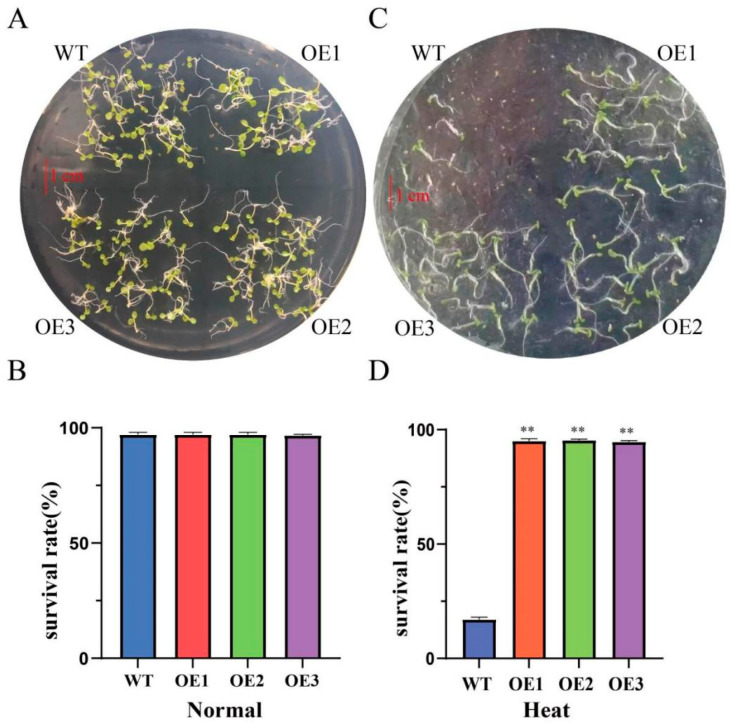
*ClaHSP17.4*-OE lines enhance tolerance to heat stress during the germination stage in *Arabidopsis*. (**A**,**B**) Phenotype and survival rates of wild-type and *ClaHSP17.4*-OE lines under normal conditions; (**C**,**D**) Phenotype and survival rates of wild-type and *ClaHSP17.4*-OE lines under heat stress (38 °C). Note: **: significant difference, *p* ≤ 0.01.

**Figure 5 cimb-47-00606-f005:**
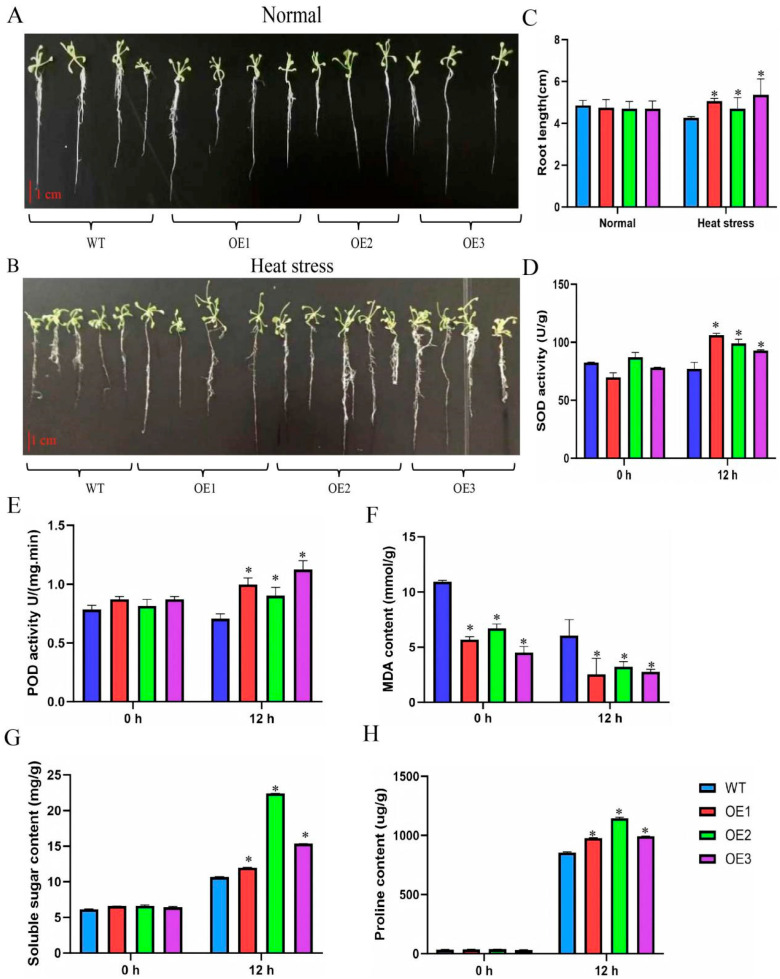
*ClaHSP17.4*-OE lines enhance tolerance to heat stress during seedling stage in *Arabidopsis*. (**A**,**B**) Phenotype under normal/under heat stress conditions of wild-type and *ClaHSP17.4*-OE seedlings; (**C**–**H**) Root length, SOD activity, POD activity, MDA content, soluble sugar, and proline content in wild-type and *ClaHSP17.4*-OE seedlings treated with heat stress at 38 °C for 12 h. Plants continually grown at 22 °C were used as controls. Bars show standard deviation of three biological replicates. Note: *: significant difference, *p* ≤ 0.05.

**Figure 6 cimb-47-00606-f006:**
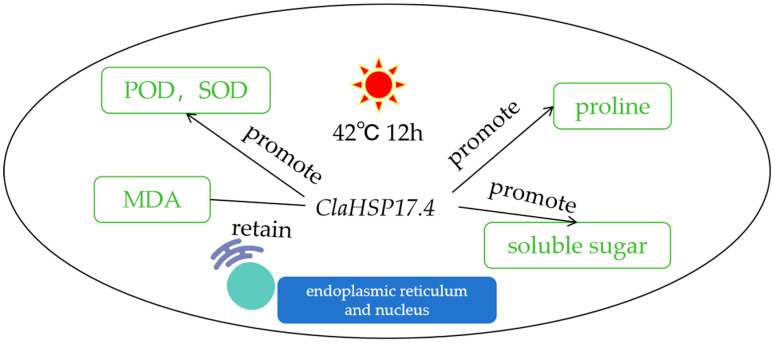
The regulation mechanisms of *ClaHSP17.4.* SOD: superoxide dismutase; POD: peroxidase; MDH: mitochondrial malate dehydrogenase.

**Table 1 cimb-47-00606-t001:** Primer sequences used for quantitative real-time PCR analysis and identification of positive plants.

Name	Forward Primer (5′→3′)	Reverse Primer (5′→3′)
qPCR-ClaHSP17.4	CGGGGTACCATGGATCTCAGAATCATGG	CGCGGATCCTTGACCTTGACCTCAACG
qRT-PCR-ClaHSP17.4	GAGGGAGGAGGAGAAAGAG	CCTGACAAACCGCTGATAT
β-actin (Cla007792)	CCATGTATGTTGCCATCCAG	GGATAGCATGGGGTAGAGCA

**Table 2 cimb-47-00606-t002:** Prediction of cis-acting elements in promoter region of *ClaHSP17.4* gene.

Component Name	Function	Quantity
Box 4, TCT-motif, AE-box	part of light responsiveness	3
ARE	anaerobic induction	4
O2-site	zein metabolism regulation	1
MYB, Myb-binding site, Myb	MYB	7
TGACG-motif, CGTCA-motif	MeJA responsiveness	2
WRE3	WRE3	1
TGA-element	auxin-responsive element	1

## Data Availability

All datasets generated for this study are included in the article.

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
