# Peer review of "Heterologous Watermelon HSP17.4 Expression Confers Improved Heat Tolerance to Arabidopsis thaliana"

_cimb, 2025, doi:10.3390/cimb47080606_

Round 1
Reviewer 1 Report
Comments and Suggestions for Authors
- In the section of Introduction, the research progress of the impact of high temperature on watermelon industry and watermelon HSP are added;
- Need to add the difference between heat sensitive and heat resistant watermelon HSP17.4 genes;
- Watermelon takes fruit as the main product, and transgenic should be mainly verified by watermelon, strawberry or tomato, focusing on the effect of the gene on fruit formation;
- In the section of 2.1, “All materials immediately frozen in liquid nitrogen” , the “materials” is leave or petioles, or roots?
- Line 171, “OD600” should be “OD600”.
6 Line 178, “mg/L” should be “mg·L-1”, and the same below, such as line 213 “μmol/m2/s”
- When transgenic into Arabidopsis, the gene detection should be in Arabidopsis plants, including qRT-PCR and qPCR. Therefore, the detection , qRT-PCR and qPCR primers in transgenic Arabidopsis should be increased.
- Lines 234-235, 410, 471 and 477, “错误!未找到引用源“
- In the section 2.6, should be moved to 2.2, and the 2.2 may be the gene obtain, bioinformatics analysis based on Results order.
- Figures in this manuscript should be enlarge and clear (including font, legend), such as Figure 1, Figure 5B/5D. The Figure 5A/C, Figure 6A/B should added the scale.
- Added the HSP17.4 genes expression levels in ClaHSP17.4-overexpressing Arabidopsis lines.
- In the sections Discussion and Conclusion should be refine.
- In the section of Introduction, the research progress of the impact of high temperature on watermelon industry and watermelon HSP are added;
- Need to add the difference between heat sensitive and heat resistant watermelon HSP17.4 genes;
- Watermelon takes fruit as the main product, and transgenic should be mainly verified by watermelon, strawberry or tomato, focusing on the effect of the gene on fruit formation;
- In the section of 2.1, “All materials immediately frozen in liquid nitrogen” , the “materials” is leave or petioles, or roots?
- Line 171, “OD600” should be “OD600”.
6 Line 178, “mg/L” should be “mg·L-1”, and the same below, such as line 213 “μmol/m2/s”
- When transgenic into Arabidopsis, the gene detection should be in Arabidopsis plants, including qRT-PCR and qPCR. Therefore, the detection , qRT-PCR and qPCR primers in transgenic Arabidopsis should be increased.
- Lines 234-235, 410, 471 and 477, “错误!未找到引用源“
- In the section 2.6, should be moved to 2.2, and the 2.2 may be the gene obtain, bioinformatics analysis based on Results order.
- Figures in this manuscript should be enlarge and clear (including font, legend), such as Figure 1, Figure 5B/5D. The Figure 5A/C, Figure 6A/B should added the scale.
- Added the HSP17.4 genes expression levels in ClaHSP17.4-overexpressing Arabidopsis lines.
- In the sections Discussion and Conclusion should be refine.
Reviewer 2 Report
Comments and Suggestions for Authors
Abstract is well structured and contain relevant novel information for readers.
Introduction: At the later part of the introduction section, authors can highlight the shortfall or limitation of the study, to provide direction for future research into this domain or topic.
Materials and methods
2.1.
- Authors should provide reference or data to support why the chose to study this particular variety of watermelon than other known varieties. What makes this variety unique?
- What is the nutrient composition of the custom nutrient substrate?
- Information about the reason to perform treatment on 3-weeks old watermelon should be provided. The reason for selecting 42oC for heat treatment and the chosen sampling time, 1, 4, and 12 h should also be supported with references or data.
- “each treatment included three times’ needs clarification.
- State the experimental approach used. Was it CRD, RCBD, etc.
Discussion
- Focus the discussion section on the physiological and molecular implication of the findings. The current form seems more of a reporting. I believe a clear explanation based on the physiological significance will enhance the quality of the manuscript.
- Please check and delete the Chinese characters in the discussion section.
- The relationship between osmoregulation and the expression of the putative genes can be enhanced with Pearson’s correlation analysis or regression analysis.
- The regulation of osmolyte and their contribution to the study should be clearly explained.
- Authors should check for few grammatical and typographical errors.
Conclusion
- The conclusion section can be enhanced with a model or flow chart to highlight the key findings of the study.
- I will suggest to authors to provide recommendation about the limitations of the study for future research direction.
Round 2
Reviewer 1 Report
Comments and Suggestions for Authors
The manuscript has been appropriately revised and is recommended for acceptance.
Comments on the Quality of English LanguageThe manuscript has been appropriately revised and is recommended for acceptance.